# An IoT-Based Data-Driven Real-Time Monitoring System for Control of Heavy Metals to Ensure Optimal Lettuce Growth in Hydroponic Set-Ups

**DOI:** 10.3390/s23010451

**Published:** 2023-01-01

**Authors:** Sambandh Bhusan Dhal, Shikhadri Mahanta, Jonathan Gumero, Nick O’Sullivan, Morayo Soetan, Julia Louis, Krishna Chaitanya Gadepally, Snehadri Mahanta, John Lusher, Stavros Kalafatis

**Affiliations:** 1Department of Electrical and Computer Engineering, Texas A&M University, College Station, TX 77843, USA; 2Department of Biological and Agricultural Engineering, Texas A&M University, College Station, TX 77843, USA; 3Department of Dairy Technology, National Dairy Research Institute, Karnal 132001, India

**Keywords:** hydroponic, Machine Learning, Linear Support Vector Machine, pairwise correlation, ExtraTreesClassifier, Xgboost, closed loop system

## Abstract

Heavy metal concentrations that must be maintained in aquaponic environments for plant growth have been a source of concern for many decades, as they cannot be completely eliminated in a commercial set-up. Our goal was to create a low-cost real-time smart sensing and actuation system for controlling heavy metal concentrations in aquaponic solutions. Our solution entails sensing the nutrient concentrations in the hydroponic solution, specifically calcium, sulfate, and phosphate, and sending them to a Machine Learning (ML) model hosted on an Android application. The ML algorithm used in this case was a Linear Support Vector Machine (Linear-SVM) trained on top three nutrient predictors chosen after applying a pipeline of Feature Selection methods namely a pairwise correlation matrix, ExtraTreesClassifier and Xgboost classifier on a dataset recorded from three aquaponic farms from South-East Texas. The ML algorithm was then hosted on a cloud platform which would then output the maximum tolerable levels of iron, copper and zinc in real time using the concentration of phosphorus, calcium and sulfur as inputs and would be controlled using an array of dispensing and detecting equipments in a closed loop system.

## 1. Introduction

Hydroponics and aquaponics-based systems have been employed as a viable alternative to traditional agricultural practices during the last several decades since they require 50 to 70% less water due to the system’s recyclability [1,2,3]. Furthermore, they have demonstrated tremendous efficiency in that they require less pest control and are less influenced by adverse climatic weather conditions, resulting in an increase in produce output. As we move towards a digital agricultural future, attempts are being made to build smart IoT-based hydroponic systems for commercial implementation.

Recent research in the field of alternative agriculture have used predictive methodologies to boost crop yield in a sustainable manner. Some of the studies have been mentioned as follows. The sensor data in the greenhouse recorded in an aquaponic system, and the fish count retrieved using the R-CNN algorithm were sent into the AutoML algorithm to activate the actuators required to manage the ambient conditions [4]. Dhal et al. devised a ML-based smart IoT system for measuring and adjusting ammonium and calcium concentrations dependent on season utilizing a feedback loop to achieve sustainable development of fish and plants in a single setup [5]. Similar studies have been carried out on using ML techniques for selecting the most optimal nutrients to be regulated in aquaponic environments through a pipeline of feature selection techniques for symbiotic growth of fish and plants in coupled environments [6,7]. Comparative research of ML-based techniques using images of lettuce was undertaken in [8] to examine what illnesses can be acquired throughout the growing phase. Hiram Ponce et al. in [9] and Yadav et al. in [10] used Deep Convolutional Neural Networks for feature extraction on leaf pictures to identify nutrient shortage in tomatoes and foliar disease in apple trees, respectively. Having said that, relatively few studies have been undertaken on the influence of heavy metals on crop development in a hydroponic settings, which is the primary goal of our research.

Some of the studies emphasizing on the uptake of heavy metals by plants in hydroponic solution have been discussed below. The impact of biodegradable chelating agents SS-EDDS on heavy metal absorption when growing sunflowers in hydroponic solution was researched in [11], and it was determined that it improved the uptake of non-essential metals from the solution. On a similar vein, Mahanta et al. [12] did a study for growing soybeans in hydroponic solution and discovered that seeds treated with plasma activated water had a significantly lower absorption of heavy metals than seeds treated with tap water. Michalska et al. investigated the influence of lead and cadmium on the growth of three lettuce variations in hydroponic culture and how it affected the absorption of macro and micronutrients in their root and shoot tissues [13]. Furthermore, Peralta-Videa et al. conducted research on the absorption of ambient heavy metals by plants as well as the harmful effects of arsenic, cadmium, and chromium on the human body [14].

Stating the above, the major objective of our research was to create a Machine-Learning-based IoT system that can output the heavy metal concentrations that can be tolerated in a commercial hydroponic system for maximum lettuce growth based on nutrient concentrations monitored in real-time. The prototype’s fundamental feature was to sense and adjust nutrient parameters using a closed-loop system to keep heavy metal concentrations in the hydroponic solution below allowable levels.

## 2. Materials and Methods

While laboratory setups can test the nutritional profiles of water samples at specific quantities, the processing time varies from few hours to a few weeks depending on the queue size. The proposed design would be a useful tool for quickly determining the concentration of the observed hydroponic environment. When enabled, the Nutrient Monitoring System (NMS) would monitor calcium, phosphate, and sulfate concentrations. Each sample would accurately measure and emit a signal exactly proportionate to the concentration of the chemical. The primary control unit, or microcontroller, would receive the spectrophotometer’s calculations and generate a signal, which would then be transformed into human-readable data points and saved in the database. A pre-trained ML model stored on the database is then run on this collection of data, and the maximum tolerated heavy metal concentrations are shown through an Android User Interface based on the ML model’s output.

A prototype of the block diagram of the NMS is as shown in Figure 1.

The subsystems mentioned in the functional prototype of the NMS have been described in the sections stated below.

### 2.1. Spectrophotometer System

The spectrophotometer system is made up of a light source that goes through a monochromator, which is a device that divides light into separate wavelengths, and an aperture. The resolution of the light flowing through is altered by the aperture. The light then passes through the sample and is absorbed by the source. The light that reaches the detector is that which was not absorbed. A schematic of the spectrophotometer has been shown in Figure 2.

In order to analyze the performance of the spectrophotometer, a calibration table was created in order to quantify the concentration of the unknown material. To construct a line of best fit, the calibration curve required samples with pre-defined concentration levels. The instrument analyzed the absorbance values of each concentrated sample using a pre-defined wavelength (610 nm in Table 1). This resulted in a set of data points that were plotted with a line of best fit to form a linear equation. With the y-value denoting the absorbance and the x-values denoting the respective concentrations, this equation was used to calculate the absorbance of the unknown test samples.

### 2.2. Positioning and Dispensing System

The positioning and dispensing system was an essential component of the NMS, allowing for the automation of water filling, reagent filling and cuvette placement in a spectrophotometer, minimizing the amount of hands-on work required. The details of the positioning and dispensing system has been shown in Figure 3.

In this system, each stage of the sequence was performed by one Nema 17 bipolar stepper motor, two valve solenoids, one A4988 motor driver, and one ESP32 microcontroller. A stepper motor was utilized to spin an arm holding a cuvette across the various phases of the conveyor system. The dispensing system used a water pump to provide both water and a reagent through a 2 mL tube. When the upstream valve solenoid was shut down, the water and reagent combination was trapped in the tube. The downstream valve solenoid then opened, allowing the water and reagents to enter the cuvette. A motor driver received 5 V from an ESP32 microprocessor and 12 V from the power management system to power the stepper motor. Two digital pins were employed in the motor driver to set the stepping type and direction of each motor. The circuit required a PN2222 transistor, a 1N4001 diode, and a 250 resistor for the water pump. A sequence of instructions was written onto the ESP32 microcontroller using the Arduino integrated development environment (IDE) to accomplish duties given to the positioning and dispensing system.

This system primarily consisted of two main steps: an AC-DC conversion system which steps down the voltage from 120 V AC supply to a 24 V DC supply, and a buck converter which takes an input of 24 V from the AC-DC converter and gives two stable output voltages of 12 V and 5 V, respectively. The entire schematic of the power management system has been depicted in Figure 4.

From Figure 4, it can be observed that the input 120 V AC was reduced by the transformer to an output 24 V AC. A bridge rectifier was used to complete this scheme. Direct current was produced by the bridge rectifier from the alternating current at the transformer’s output. The bridge rectifier used had dual outputs, one of which was grounded, while the other carried 24 V AC to the next section of the circuit, which is the buck converter system. In this case, the buck converter used was a simple step-down DC-DC converter capable of supplying a stable voltage given a wide range of input voltage. The two buck converters were slightly different in the fact that one will step down the voltage to 12 V while the other will supply 5 V. The capacitor values, inductor values, and diodes were chosen through calculations to produce appropriate output voltage as well as currents, found in the LM2596 datasheet. These components were all chosen for the specific voltage output as well as the current that passes through the circuit. With all these components, there were two final outputs giving a stable 12 V and 5 V supply to the electrical components in the system.

### 2.3. Database and Machine Learning System

The purpose of the system was to collect data from the spectrophotometer via Wi–Fi on the ESP32 microcontroller, and then send and store that information within a database hosted on Google Firebase. The pre-trained ML model hosted on the Firebase is run on the calcium, phosphate and sulfate concentrations which are selected through a pipeline of feature selection techniques on a dataset recorded over the course of a year from three hydroponic farms in East-Central Texas.

### 2.4. Android Application System

The Android phone application acted as a portable display, displaying the relevant heavy metal concentration value, sorted by measurement date, based on the output of the ML model hosted on Google Firebase. The spectrophotometer data was communicated to the Firebase (database) through a Wi-Fi connection on the ESP32 module. When the output of ML model hosted on Google Firebase was 0, the android phone application displayed concentrations of iron, copper and zinc concentrations that could be tolerated in the hydroponic environment to be 0.03 ppm, 0.006 ppm and 0.06 ppm, respectively. On the other hand, when the output of the Firebase ML model was 1, the android phone application displayed concentrations of the iron, copper and zinc amounts that could be allowed to be 2.04 ppm, 0.172 ppm and 0.54 ppm, respectively.

## 3. Results

The water samples were collected weekly from these farms and forwarded to Texas A&M University’s Soil, Forage, and Water Testing Laboratory for nutrient profiling. Calcium, magnesium, sodium, potassium, boron, carbonate, bicarbonate, sulfate, chloride, nitrate, phosphate, iron, and copper quantities (all measured in ppm) were extracted from each sample and appended to the dataset. The final dataset, which we utilized in our example to conduct the preliminary analysis, had 238 observations and 14 predictors.

The iron, copper and zinc concentrations were considered as response variables, and the remaining 12 predictors were utilized to carry out the analysis. We began by considering the entire dataset as unsupervised and used K-Means clustering with the value of K set to 3. Out of the 238 observations included in the study, 116 were categorized as Class 0 and 122 were classified as Class 1. Asthe dataset was sparse and high-dimensional, a pipeline of feature selection techniques comprising of Pairwise correlation matrix, XGBoost classifier, and ExtraTreesClassifier was run on it to determine the most relevant predictors in the study.

From Figure 5, calcium, magnesium, sodium, phosphates, boron and sulfate were chosen as the top 6 predictors for analysis as they had feature importance over 200. XGBoost was used as it uses Gini Impurity to rank features in order of their importance using an ensemble of decision trees, thus making it a robust approach for feature selection. The rest of the predictors were comparatively less important in our resulting analysis, due to which, it was decided to eliminate them before proceeding with the application of ExtraTreesClassifier for selection of the top three predictors which were regulated at regular intervals using an IoT based set-up.

In this case, ExtraTreesClassifier was used as it was an ensemble learning method fundamentally based on decision trees, which would randomize the decisions and data subsets to minimize overfitting owing to its property of being a low variance estimator. This has been stated in Figure 6 where the feature importance of calcium, phosphate, and sulfate were found to be 34%, 27%, and 18%, respectively, accounting for 79% of the overall feature importance in the dataset. As a result, the values of the attributes were employed as predictors in the ML model to generate the heavy metal concentrations that may be tolerated in a hydroponic setup based on the ML output.

As previously stated, the entire dataset was treated as a binary classification problem, and based on the historical values of the dataset, a specific value of iron, copper and zinc that can be tolerated in a hydroponic set-up for optimal lettuce growth was prescribed based on the value of the ML classifier’s output. On the dataset, a 5-fold Cross-Validation with 15 repetitions was done to create the aggregate testing accuracy and pick the best classifier. Figure 7 shows the testing accuracy of each of the ML classifiers.

Figure 7 showed that out of the three classifiers employed on the dataset, Linear SVM outperformed the others. Linear SVM proved to be effective in this case where the dimensionality of the data was large and was comparable to the number of samples in the dataset. The other significant advantage over the other classifiers was that the classifier chose the decision boundary that maximized the distance from the nearest point of both the classes. Therefore, from Figure 7, it can be observed that when the penalty parameter was set to 10, the greatest aggregate testing accuracy was observed to be 75% in the instance of Linear SVM. As a result, it was decided to use the above-mentioned classifier in the analysis.

A set of maximum tolerated levels of iron, copper and zinc in the hydroponic setup were recommended based on the output of the Linear SVM using the concentrations of calcium, phosphate, and sulfate as inputs to the ML model. The training dataset was used to get the median value of these heavy metal concentrations. When the result of the ML model was class 0, the iron, copper and zinc concentrations that could be tolerated in the hydroponic environment were 0.03 ppm, 0.006 ppm and 0.06 ppm, respectively. Similarly, when the ML model output was 1, the iron, copper and zinc amounts that could be allowed were 2.04 ppm, 0.172 ppm and 0.54 ppm, respectively.

## 4. System Performance

The recommended calcium, sulfate, and phosphate concentrations to be maintained for maximum lettuce development in a hydroponic setup were stated as 130 ppm, 125 ppm, and 25 ppm, respectively. To maintain this concentration in a 450 L hydroponic set-up, 90 g of calcium sulfate powder and 31 g of magnesium phosphate powder were manually supplied to the hydroponic set-up.

Following that, the constructed spectrophotometer system was utilized to measure real-time values of these nutrients, which were then stored on Google Firebase, where the ML model was located. Based on the ML result, the suitable iron and copper values that may be tolerated in a hydroponic setup were shown via an Android application interface.

The spectrophotometer system also had a link to the Nutrient Dispensing System, which had two cuvettes with a dispensing capacity of 2 mL each. Both systems were linked in a closed loop via a feedback loop, with the spectrophotometer system sensing the nutrient parameters in the solution and sending the readings to the dispensing system, which released the nutrient solution if the measured concentrations were less than the recommended parameters. The sensing and dispensing devices were operated five times for a single run of the NMS, and the average value of the nutrient concentrations recorded from the spectrophotometer was saved on the Firebase and was later used as input to the ML model.

To quantify our findings, during a single growing season of lettuce (21 days), multiple test runs were carried out to test the heavy metal concentrations in the hydroponic solution, in periodic intervals, and have been stated in Table 2.

From Table 2, it can be stated that as we move through the growth period of lettuce, the required nutrient concentration of calcium, phosphate and sulfate were altered according to the requirements of the crop. The corresponding heavy metal concentrations in the hydroponic solution was observed and it can be stated that the iron, copper and zinc concentrations stayed within the maximum permissible limits as prescribed by the ML algorithm, ensuring optimal growth of lettuce in the hydroponic set-up.

## 5. Discussion

The impact of calcium and sulfate addition was researched in [15], and it was determined that the dry matter yields of majority of the plants under high calcium sulfate treatments were higher than those under low treatment. Many research studies have also shown that adding calcium to the hydroponic solution raised the pH of the solution. The influence of pH on copper absorption by plants was examined in [16], and it was found that increasing the pH of the soil lowered the rate of copper absorption by plants as measured by shoot biomass and root elongation, lowering the likelihood of copper poisoning.

According to research on the effect of copper toxicity in [17], when the concentration of copper was above a particular threshold, which in this case was 100 micro-Moles, the relative growth rate dropped and severe browning of the leaves occurred, culminating to necrosis. The effect of increasing copper content on phosphorus absorption by plants in hydroponic media was explored in [18,19], where severe imbalances in the nutritional values of the plants as well as inhibited plant development were reported.

Similarly, the influence of pH on iron absorption by plants was investigated. According to [20], the pH of the soil should be kept at a reasonable level for ferric iron to be liberated from ferric oxides and become more accessible for absorption by plant roots. Similarly, a lower pH level meant that plants absorbed more iron, which can contribute to iron poisoning. In [21], a full analysis of the hazardous quantities of iron that resulted in young plants suffering from elevated oxidative stress was explored with lower relative growth rates.

The negative effect of iron on phosphate concentrations in growth contexts was explored in [22], where phosphate deficit caused by high iron inhibited primary root growth and delayed lateral root development. This discovery was repeated in [23], which indicated that growing barley in a phosphorus-depleted hydroponic medium led in iron plaque development in the root system.

Similarly, the negative effects of zinc on the growth of plants in hydroponic environments have been studied where excess of zinc uptake by the plants have shown inward-rolled leaf edges and a damaged and brownish root system with short lateral roots. They have also shown a decrease in the uptake of the essential nutrients by the plants resulting in de-epoxidation of violaxanthin cycle pigments [24,25]. A high concentration of zinc in the hydroponic solution has also shown to result in the formation of zinc-phosphorus complex formation in the roots of the plants, resulting in inhibition of phosphorus movement which is an essential component for plant growth in a hydroponic system [26]. As a more alkaline pH level in the growth media would allow more absorption of zinc by the roots, it was advised to balance the nutrients in a manner that the overall pH of the solution would be low in order to prevent zinc toxicity [27].

As a result of the above, the main objective of the research was to build a system to balance the pH of the hydroponic solution by maintaining appropriate concentrations of calcium, sulfate and phosphate so that the uptake of heavy metals by the plants was minimized, thus resulting in higher yield.

## 6. Conclusions and Future Work

In conclusion, a smart IoT-based NMS was designed to take calcium, phosphorus and sulfate concentrations as inputs which were selected as features to be regulated through a pipeline of feature selection techniques like the pairwise correlation matrix, Xgboost classifier and ExtraTreesClassifier. The detected nutrient concentrations were passed on to a binary classifier which in this case is a Linear SVM (penalty parameter = 10) as the testing accuracy was around 75%. Depending on the value of the output from the ML classifier, if the output was 0, the maximum tolerable levels of iron, copper and zinc in the hydroponic solution was 0.03 ppm, 0.006 ppm and 0.06 ppm, respectively. Similarly, if the output from the ML classifier was 1, the recommended maximum tolerable levels of iron, copper and zinc were specified to be 2.04 ppm, 0.172 ppm and 0.54 ppm. This was verified experimentally at regular intervals for one growth cycle of lettuce (21 days) in a 450 L hydroponic test set-up where the heavy metal concentrations were monitored accurately for different concentrations of nutrients as inputs to the ML model.

In this case, data has been recorded from three aquaponic farms in South-East Texas, which has a more humid sub-tropical climate. In the future, data from more geographically varied terrains will be collected to generate a dataset with greater variance in data, which will aid in the development of a more dependable and robust ML model.

The spectrophotometer, which presently only measures three nutrients, i.e., phosphorus, calcium and sulphur could be expanded to monitor additional chemical characteristics of the hydroponic solution.

The present size of the positioning and dispensing system, which comprises of two dispensing units, may be enlarged to create a bigger system that can be utilized in larger commercial setups. Furthermore, the existing proposed system lacks a method to monitor or adjust heavy metal concentrations, i.e., iron, copper and zinc in real time, which can be integrated.

## Figures and Tables

**Figure 1 sensors-23-00451-f001:**
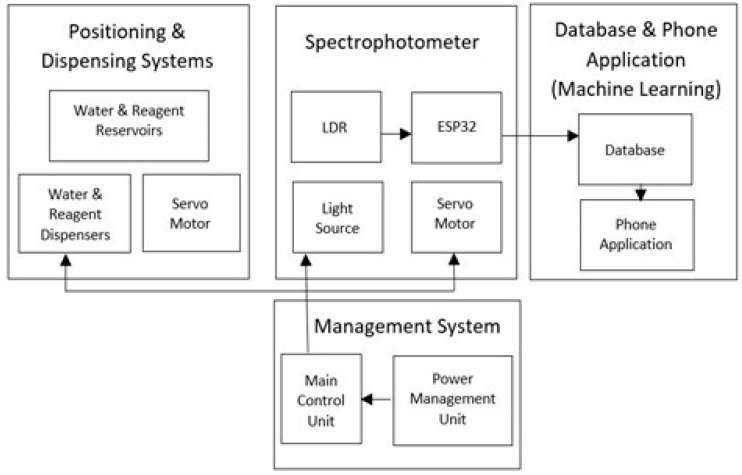
Functional Prototype of the NMS.

**Figure 2 sensors-23-00451-f002:**
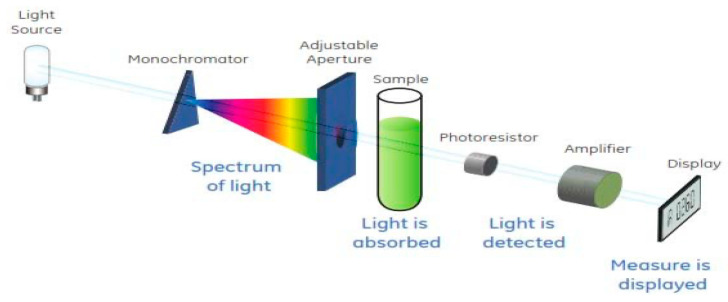
Schematic of the Spectrophotometer set-up.

**Figure 3 sensors-23-00451-f003:**
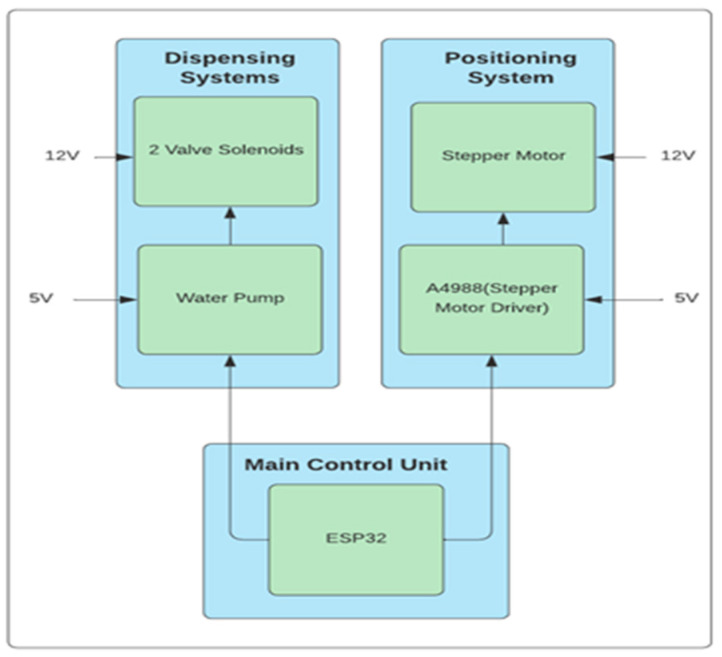
Schematic of the Positioning and Dispensing system of the NMS.

**Figure 4 sensors-23-00451-f004:**
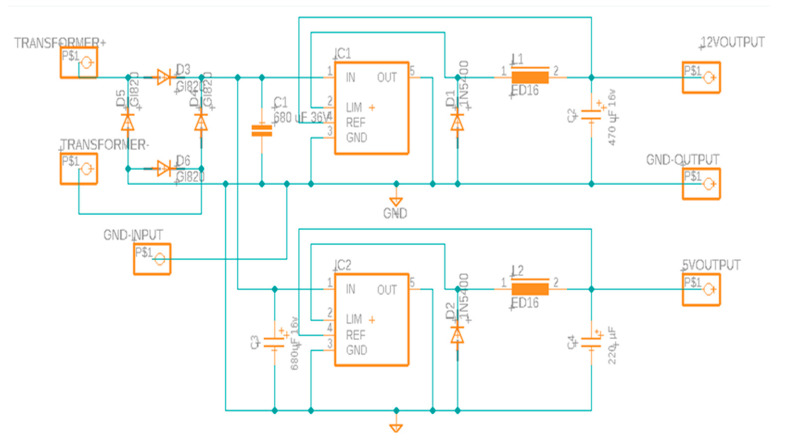
Power management circuit for the Dispensing and Positioning system of the NMS.

**Figure 5 sensors-23-00451-f005:**
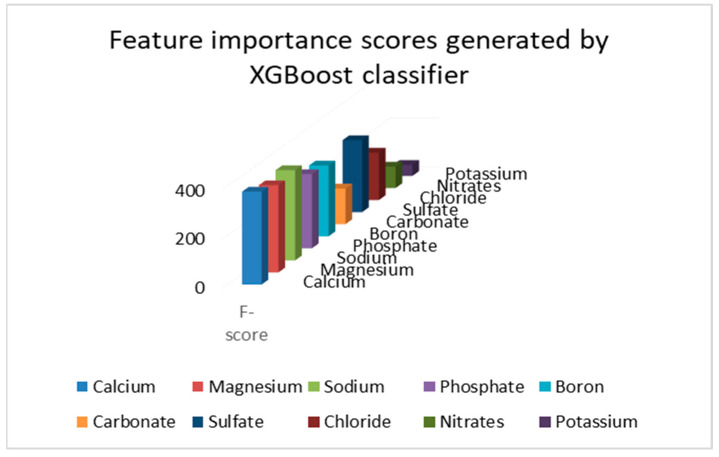
F-scores generated by the XGBoost classifier.

**Figure 6 sensors-23-00451-f006:**
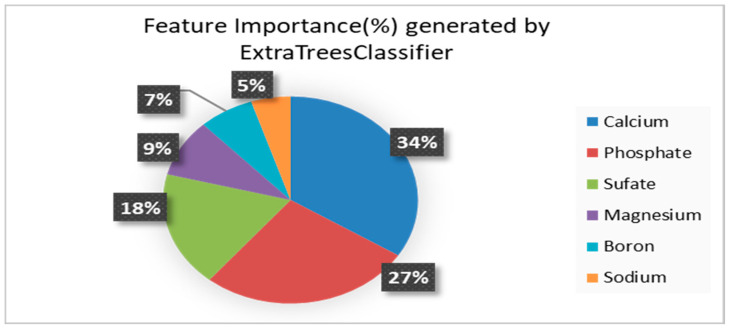
Feature Importance of the predictors generated by the ExtraTreesClassifier.

**Figure 7 sensors-23-00451-f007:**
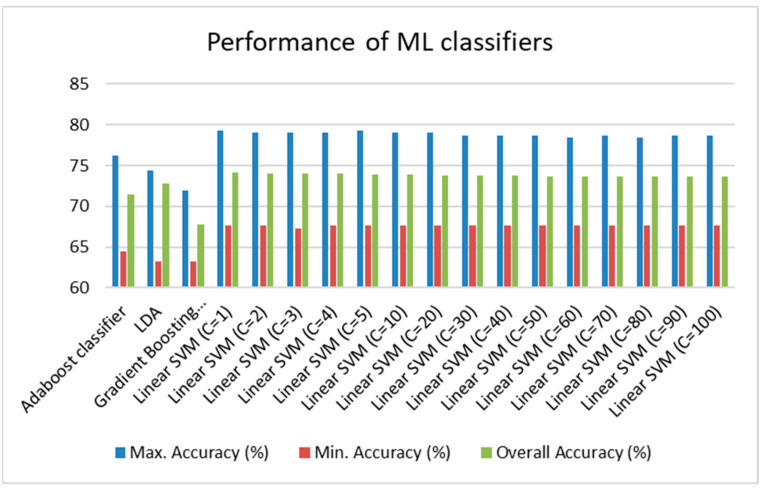
Maximum, minimum and overall testing accuracy of the ML classifiers.

**Table 1 sensors-23-00451-t001:** Concentration calibration data for Calcium, Phosphate and Sulfate.

SAMPLE ID	STD0	STD1	STD2	STD3	STD4	STD5	UNKNOWN
TYPE	Standard	Standard	Standard	Standard	Standard	Standard	Standard
CALCIUM							
CONCENTRATION	0.000	28.800	57.600	86.400	115.200	144.000	
WL610.0	0.000	0.210	0.266	0.306	0.358	0.493	
WEIGHT FACTOR	1.000	1.000	1.000	1.000	1.000	1.000	
PHOSPHATE							
CONCENTRATION	0.000	0.516	1.032	1.550	2.060	2.580	
WL610.0	0.000	0.226	0.359	0.478	0.583	0.847	
WEIGHT FACTOR	1.000	1.000	1.000	1.000	1.000	1.000	
SULFATE							
CONCENTRATION	0.000	4.800	9.640	14.460	19.280	24.100	10.000
WL450.0	0.000	0.429	0.753	1.091	1.086	1.432	0.638
WEIGHT FACTOR	1.000	1.000	1.000	1.000	1.000	1.000	1.000

**Table 2 sensors-23-00451-t002:** Set-Up to monitor heavy metals in a test hydroponic set-up by regulation of nutrients.

Sl. No.	Date	Nutrient Concentrations (ppm)			Class	Heavy Metal Concentrations (ppm)		
		Calcium	Phosphate	Sulfate		Iron	Copper	Zinc
1	1 October 2022	15	7	7	0	0.005	0.005	0.04
2	5 October 2022	12	6.5	160	1	0.8	0.77	0.35
3	13 October 2022	95	5.4	166	1	0.65	0.7	0.28
4	18 October 2022	3	4.27	7	0	0.02	0.003	0.03
5	22 October 2022	84	5.36	216	1	1.8	0.13	0.48

## Data Availability

https://zenodo.org/record/7317974#.Y3H_lXbMJD8 (accessed on 14 November 2022.)

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
