# Peer review of "An IoT-Based Data-Driven Real-Time Monitoring System for Control of Heavy Metals to Ensure Optimal Lettuce Growth in Hydroponic Set-Ups"

_sensors, 2023, doi:10.3390/s23010451_

Round 1

Reviewer 1 Report

The current research problem identified is clear & unique, with explicit solution provided using the IoT driven real time data. The methodology developed is tested using a closed loop system. The machine learning model is experimentally verified with future work to be improved further, which paves way for upcoming researchers.

Author Response

Thank you for your comments.

Reviewer 2 Report

The work proposed is ,  a smart IoT-based nutrient monitoring system  designed to take 281 calcium, phosphorus and sulfate concentrations as inputs which are selected as features 282 to be regulated through a pipeline of feature selection techniques like the pairwise corre- 283 lation matrix, Xgboost classifier and ExtraTreesClassifier. The explanation is legible and relevant. 

The author have compared the results with adaboost, LDA, Gradient boosting, Linear SVM with different C values. The author can also compare with some other classifiers. 

Author Response

Thank you for your comments and suggestions. The reasons for using the stated classifiers have been explained in the comments below:

i. Adaboost classifier was used as it is less prone to overfitting as the input parameters are not jointly optimized.

ii. LDA was used as it uses the information from all the features to create new axes which in turn minimizes the variance and maximizes the distance between the classes.

iii. Gradient Boosting was used as it provides the best predictive accuracy, can optimize on different loss functions and provides several hyperparameter tuning options that make the function fit very flexible, eliminates data pre-processing, and helps us to handle missing data.

iv. Linear SVM was used as it is effective in cases where the number of dimensions are greater than the number of samples and is very effective in cases where the dimensionality of the dataset is large.

We did not use classification algorithms like Logistic Regression and Random Forest due to the following reasons.

i. Logistic regression is inaccurate if the sample size is too small.

ii. The main limitation of random forest is that a large number of trees can make the algorithm too slow and ineffective for real-time predictions. In general, these algorithms are fast to train, but quite slow to create predictions once they are trained.

Reviewer 3 Report

I would like to congratulate the authors for submitting their research work in this journal. The manuscript is a well written analysis with good contents. An IoT-Based Data-Driven Real-Time Monitoring System for Control of Heavy Metals to ensure optimal lettuce growth in Hydroponic Set-Ups. Still, I believe some ground work is needed before the research is suitable for acceptance. Authors are advised to update their work according to the suggestions recommended.

1.  The objective of the paper is well-defined in the abstract section. However, there is a lack of clarity about the future scope of the research from the abstract. There is a need to include a well-defined future scope of the research.

2. There are grammatical mistakes in some parts of the manuscript that need to be corrected.

3. There is a need to define each abbreviation in the paper at first appearance only and later use the abbreviation throughout the paper.

4. Use of  Radial Support Vector Machine (R-SVM) is used in the proposed model must be justified

5. Authors have mentioned a Machine Learning (ML) model hosted on an Android application but the explanation seems ambiguous, so authors are encouraged to add some more details about the impact of the Machine Learning procedure used in the study.

6. There is a need to explain Figure 5,6,7 in more detail.

7. There is a lack of clarity regarding the Power management circuit for the Dispensing and Positioning system of the NMS. So, describe it in some more detail for clarity.

Round 2

Reviewer 3 Report

Originality of the work: Acceptable

Subject Relevance: Good

Professional/Industrial Relevance: Good

Completeness of the Work: Acceptable for publication

I.  The Title "An IoT-Based Data-Driven Real-Time Monitoring System for Control of Heavy Metals to ensure optimal lettuce growth in Hydroponic Set-Ups" are investigated with all aspect of research

II.  I completely agree with incorporated comments and improved quality of the work